# Efficacy of wooden toy training in alleviating cognitive decline in elderly individuals with cognitive impairment: A cluster randomized controlled study

Qiuping Cheng[1], Hanqian Wang[2], Mengni Cui[3], Qunlong Wang[1]*, Lu Li[2]*

**1** Institute of Modern Services, Zhejiang Shuren University, Hangzhou, China, **2** Institute of Social Medicine, School of Medicine, Zhejiang University, Hangzhou, China, **3** Translational Institute Medical Science, Zhejiang Shuren University, Hangzhou, China

\* zjsrujzyl@126.com (QW); lilu@zjsru.edu.cn (LL)

**Data Availability Statement:** The final dataset is available on the figshare, DOI: 10.6084/m9.figshare.26695081. If you encounter any issues accessing the data or require further assistance,

## Abstract

### Background

With the increasing global aging population, the health and welfare of elderly individuals, especially individuals with prevalent mild cognitive impairment (MCI) living in nursing homes, have become critical concerns. These concerns highlight the urgency of developing effective interventions to address the cognitive and psychological needs of elderly individuals, ensuring their well-being and alleviating the burden on their caregivers.

### Objective

This study investigates the impact of wooden toy training on mitigating cognitive decline in elderly individuals with cognitive impairment. It evaluates how this training influences cognitive functions and psychological well-being, exploring whether it can slow or reverse the progression of cognitive decline. This outcome will be assessed in a randomized controlled trial, in which changes in cognitive ability and psychological health indicators among the participants will be measured.

### Design

A two-arm, open-label, cluster-randomized controlled trial.

### Setting(s)

The study was conducted in two nursing homes, which served as both the recruitment sites for participants and the intervention locations. These nursing homes were selected for their ability to facilitate the intervention and for their representative demographic characteristics of the elderly population. The settings provided a controlled environment that was conducive to implementing the wooden toy training program and observing its effects on the participants.

please contact the corresponding author, Qunlong Wang, at the email: zjsrujzyl@126.com.

**Funding:** This work was supported by The Planning Projects of Zhejiang Provincial Philosophy and Social Sciences (24NDJC193YB, awarded to QC); Open Fund Project of the Modern Service Research Center of Zhejiang Shuren University (SXFJZ202301, awarded to QC); and The National Social Science Fund of China (21BGL235, awarded to LL). The funders had no role in study design, data collection and analysis, decision to publish, or preparation of the manuscript.

**Competing interests:** The authors have declared that no competing interests exist.

## Participants

A total of 76 elderly participants with mild cognitive impairment but functional independence were recruited.

## Methods

Participants were randomized into an intervention group and a control group. The intervention group engaged in an 8-week wooden toy training program, whereas those in the control cohort received customary nursing care. Standardized cognitive and psychological well-being measures were used to assess improvements in cognitive performance and mental health.

## Results

Significant improvements were observed in the cognitive functions of the intervention group from a baseline score of M = 13.11 to M = 16.29 postintervention (95% CI [-4.44, -1.93]), along with reductions in depressive symptoms from a baseline score of M = 8.63 to M = 7.18 (95% CI [0.38, 2.51]). Additionally, engagement in activities with wooden toys significantly satisfied their need for competence, increasing from a baseline of M = 16.29 to M = 20 post-intervention (95% CI [-5.92, -1.51]), and relatedness, which improved from a baseline of M = 20.32 to M = 22.95 (95% CI [-4.73, -0.53]).

## Conclusions

This study underscores the potential of a wooden toy intervention in nursing homes that combines cognitive challenges with traditional cultural elements to improve cognitive functions in elderly individuals with mild cognitive impairment. Our findings suggest a novel method of promoting the cognitive and psychological health of nursing home residents through emotional comfort and social interaction.

## Introduction

The global population is currently facing an unprecedented aging crisis, with cognitive impairments, including mild cognitive impairment (MCI), emerging as a substantial public health challenge. These impairments, particularly dementia, a severe form of cognitive decline, exert considerable pressure on health care infrastructures and societal structures [1, 2]. The prevalence of MCI, which is often a precursor to dementia, increases in prevalence with age, increasing the risk for elderly individuals [3–5]. In China, an alarming trend has emerged, as approximately 20% of individuals aged 65 years and above are diagnosed with MCI [6], a statistic that has substantially increased among nursing home residents [7, 8]. This disparity may be attributed to the basic psychological needs of these individuals being more severely neglected or not adequately met [9, 10]. While not immediately debilitating, the subtle cognitive decline associated with MCI significantly affects individuals' psychological well-being, often manifesting as increased depressive symptoms and decreased treatment adherence [11].

Current strategies to delay cognitive decline include both pharmacological and nonpharmacological interventions [12–15]. A body of research suggests that pharmacological solutions have limited efficacy in restoring cognitive functions among elderly individuals, pointing to a

crucial need for alternative therapies [16–20]. In contrast, nonpharmacological approaches, including aerobic exercises and practices such as Tai Chi Chih and Baduanjin, have shown the potential to enhance cognitive abilities and alleviate depressive symptoms in individuals with MCI [21–24]. Furthermore, cognitive training activities, ranging from computer and board games to more culturally rooted games such as Mahjong, have been correlated with improvements in both cognitive performance and psychological well-being [25–28]. These engaging methodologies not only underline the significance of cognitive engagement in reducing MCI-associated risks but also raise questions about the long-term consistency of such intervention outcomes [29–31]. This observed inconsistency in intervention outcomes poses a significant challenge in the field of aging research, necessitating innovative solutions that can be effortlessly integrated into the daily lives of the elderly to promote sustained increases in cognitive function and overall quality of life.

In this context, wooden toys represent a unique and culturally resonant intervention. Their natural texture and inherent warmth serve not only as a source of cognitive stimulation but also as a means of emotional comfort and social interaction, which are pivotal for the cognitive health care and psychological well-being of elderly individuals. Due to their simplicity, ease of play, low cost, and affinity, wooden toys are often more easily incorporated into the daily exercise and activities of elderly individuals, especially for those who may feel uncomfortable or resistant to high-tech devices. Engaging with wooden toys allows individuals to experience a direct connection with natural materials. The tactile qualities, warmth, and heft of wood provide a rich sensory experience, potentially fostering a state of tranquility and relaxation [32–35]. In this state, the brain may release increased levels of calming neurotransmitters such as serotonin and endorphins, contributing to the alleviation of anxiety and stress [28, 36, 37]. Crucially, when these activities are conducted in group settings, they encourage social interaction and collaboration, mitigating feelings of loneliness and enhancing the emotional and social well-being of older adults [38, 39].

Moreover, quilting and similar crafting activities not only promote the maintenance and development of fine motor skills [40] but also stimulate key brain areas involved in coordinating physical movements and cognitive processes, such as the cerebellum and basal ganglia, which play vital roles in motor coordination and the acquisition of new skills [41, 42]. Hands-on manipulation of wooden toys seems to likely have a similar effect. Certain puzzle-based wooden toys, such as the "Huarong Path Slide Puzzle" require the rearrangement of various pieces to solve spatial challenges, potentially activating the prefrontal cortex and parietal lobes associated with problem-solving and spatial cognition. Prolonged engagement with such games could promote neural plasticity, slowing age-related cognitive decline [43, 44]. The diversity and interactivity of wooden toys allow elderly individuals to make choices based on their interests and capabilities. This self-directed engagement enhances their sense of agency and decision-making freedom [45] which impacts overall satisfaction and psychological well-being. Solving puzzles and overcoming challenges instill a sense of competence, bolstering their confidence and self-efficacy. In summary, wooden toys provide elderly people with multifaceted cognitive benefits, enhancing mental agility and problem-solving abilities while also fulfilling basic psychological needs and fostering social interactions that increase their quality of life. This holistic approach underscores the importance of integrative health interventions in maintaining the cognitive and emotional health of older populations.

Despite this promising premise, the correlation between such sensory-rich interactions and enhanced cognitive function and psychological well-being in the elderly is an underexplored territory. This study proposes a rigorous investigation into the efficacy of wooden toys, employing a two-arm open-label randomized controlled trial designed to yield robust scientific support for this nonpharmacological approach. In the context of an aging global

population, particularly in nursing home settings, prioritizing the cognitive and psychological health of elderly individuals is imperative [46]. In such environments, elderly individuals frequently face substantial cognitive challenges, highlighting the necessity of effective interventions. This study investigated practical alternative methods for enhancing the cognitive functions and psychological well-being of nursing home residents. These methods not only aid in mitigating the impact of cognitive decline, thereby improving the quality and effectiveness of elderly care in nursing homes, but also have the potential to benefit societal welfare.

## Methods

This study was a two-arm, open-label, cluster randomized controlled trial conducted among elderly participants with cognitive impairment but functional independence. The participants in the intervention cohort underwent wooden toy training, whereas those in the control cohort received customary nursing care throughout the study. The study received ethical approval from the Ethical Review Committee of Zhejiang Shuren University, with approval number [S23-236-001_2023SK030]. We ensured that all procedures performed in this study were conducted in accordance with the ethical standards of the institutional research committee and with the 1964 Declaration of Helsinki.

This trial was registered at the WHO International Clinical Trial Registry Platform via the Chinese Clinical Trial Registry (ChiCTR), registration number ChiCTR2400081522 on 04 March 2024. This registration confirms that all necessary details regarding the trial's objectives, interventions, and methodologies of the trial are publicly accessible, ensuring transparency and adherence to international standards for conducting clinical research.

Informed consent was obtained from all individual participants involved in the study. We have included in our manuscript a statement confirming that informed consent for participation in the study as well as for the publication of identifying information/images was obtained from all participants. Additionally, we have taken care to remove all HIPAA identifiers and other identifying information from all sections of the article, including supplementary information, to protect the privacy of the participants. We have not used colored bars/shapes for anonymization and have ensured that any identifying images/video/details for which specific permission was not obtained have been removed from the manuscript.

### Participants

Participants were enrolled using precise inclusion and exclusion criteria, ensuring the selection of individuals who were cognitively impaired yet physically functional. G-power software [47] facilitated the determination of a sample size robust enough to detect a medium-to-large intervention effect with an α of 0.05 and a power of 80%. A cohort of 34 participants for each study arm was deemed necessary. We decided to recruit an additional 4 participants to account for potential sample attrition, resulting in 38 participants for each arm.

The recruitment process, spanning from 23/5/2023 to 10/6/2023, involved administering questionnaires to prospective participants at two nursing homes in Yunhe Country, Zhejiang Province. A cluster randomized controlled trial (CRCT) design [48] was subsequently adopted, with nursing homes serving as the units for randomization, thus streamlining the logistical execution of the intervention.

The eligibility criteria for study participation included the following:

a. Aged 65 years or older with retained daily functional independence and no severe impairments in hearing, vision, or communication;

b. A Montreal Cognitive Assessment (MoCA) score of less than 26;

c. A GDS-15 score of 8 or greater;

d. Not meeting the diagnostic standards for dementia;

e. No prior exposure or engagement with elderly wooden toys by the elderly individuals;

f. Provision of a signed informed consent form.

We excluded individuals who met the following criteria:

a. Exhibited mobility difficulties;

b. Had a history of significant physical illnesses;

c. Manifested severe cognitive decline that impinged on their ability for independent functionality.

**Informed consent process.** Before any study procedures were initiated, all potential participants were provided with a detailed explanation of the objectives, procedures, potential benefits, and possible risks of the study. This process was conducted to ensure full transparency and to uphold the ethical principles of voluntariness and informed participation. After ensuring that the participants had a comprehensive understanding of the study and after addressing any questions or concerns, participants signed a written informed consent form. This form explicitly included consent for participation in the study as well as for the publication of potentially identifying information/images. The informed consent process was a critical step in our recruitment process, ensuring that participants' rights and privacy were rigorously protected, in accordance with ethical standards and guidelines.

## Intervention program

The wooden toy training was facilitated by two seasoned nursing home staff members, each with comprehensive expertise in elder care. Both individuals underwent specialized training under the guidance of a professional wooden toy craftsman to ensure the effective delivery of the intervention. The wooden toys were meticulously curated for elderly individuals with cognitive challenges and structured around thematic activities tailored to their unique needs. The training regimen (e.g., see S1 Fig) encompassed games such as "Stacking Blocks", "Skill Ball Challenge", and "Dragon Tail Wagging", all of which were calibrated to augment spatial cognition and attention span. Activities such as "Pitching Pot" and "Ring Toss" were tailored to hone fine motor skills, whereas "Tangram" and "Huarong Path Slide Puzzle" aimed to stimulate logical reasoning and problem-solving abilities. Furthermore, tools such as the "Tower of Hanoi" and "Memory Chess" were incorporated to amplify memory retention capabilities. These engagements could be undertaken individually, in pairs, or within group settings. Three times weekly, in three distinct sessions, participants in the intervention group were encouraged to diversify their selection of wooden toys, enjoying them either alone or collaboratively. Guided by ethical considerations, the wooden toy training will be extended to the participants in the control group upon the completion of the study.

## Statistical analysis

Data collection was led by the first and second authors.

a. We adhered to the intention-to-treat (ITT) principle to maintain the robustness of our analysis when handling nonrandomized missing data [49]. We employed multiple imputation methods, recognizing their effectiveness in preserving data variance and improving the

estimation accuracy. Analyses were conducted using SPSS 27, ensuring methodological rigor with a predefined significance level of $p < 0.05$.

b. We initiated our analysis by conducting a thorough examination of baseline characteristics across clusters. This process involved the computation of descriptive statistics to capture key demographic information and clinical indicators, laying a solid foundation for understanding the context within which subsequent interventions were administered.

c. After this preliminary examination, we conducted independent sample t tests to explore potential differences between the intervention and control groups at baseline. This approach was pivotal in ensuring that any disparities observed postintervention could be attributed to treatment effects rather than preexisting imbalances between the groups. Before this analysis, assumptions including homogeneity of variances were verified.

d. We then calculated the intraclass correlation coefficient (ICC) for the cognitive scores and mental health-related indicators to assess the similarity between different individuals within the same cluster. Computing the ICC was crucial as it quantified the magnitude of the cluster effect and provided guidance for subsequent statistical analysis methods [50–52]. Considering that our study design included repeated measures, namely, multiple observations of the same subjects at different time points, we had to choose an analysis method that could accommodate this data structure.

If the ICC value was close to 0, it would indicate that the differences between individuals might outweigh the differences within clusters, potentially allowing us to use traditional repeated measures analysis methods. If the ICC value was close to 1, then we opted for a mixed-effects model for further analysis to comprehensively reflect the cluster effects in the data and account for the complexity of the repeated measures data. This approach enabled a deeper understanding of the impact of wooden toy training on elderly individuals with MCI.

## Measures

In this study, we conducted assessments at the baseline and postintervention stages, categorizing the results into two primary domains: **cognitive function and psychological well-being.**

a. Cognitive functions were assessed using the Chinese adaptation of the Montreal Cognitive Assessment (MoCA), which constitutes the study's main outcome measure. This instrument has been widely adopted because of its quick and comprehensive evaluation of various cognitive domains, including executive function, naming, attention, memory, abstract reasoning, and language abilities. It is particularly applicable for identifying MCI, with scores below 26 typically indicating potential cognitive decline [53, 54]. This 30-item scale is a reliable indicator of cognitive performance, with higher scores denoting stronger cognitive abilities. In our study, the MoCA scale showed good internal consistency, with a Cronbach's alpha of 0.818 calculated from our data.

b. Psychological well-being was intricately assessed through two prisms: emotional health and basic psychological needs satisfaction.

**Emotional health.**   This assessment delved into mental health indicators, focusing on depressive symptoms and daily emotional fluctuations. We utilized the Geriatric Depression Scale (GDS-15) for its ability to gauge depressive tendencies in elderly individuals [55, 56]. Scores range from 0 to 15, with scores exceeding 8 indicating emerging depressive symptoms, underscoring an increased risk of depression. In our study, the scale's reliability was

confirmed, as it demonstrated a Cronbach's α of 0.82. Similarly, the Positive and Negative Affective Schedule (PANAS), developed by Watson, Clark, and Tellegen [57], encompasses 32 adjectives, each rated on a 5-point Likert scale, to discern the nuances of emotional experiences. It registers the intensity of positive and negative emotions, with high scores indicating prevalent positive or distressing emotions, respectively. The internal consistency reliability (Cronbach's α coefficients) of the scale in our study was 0.94 overall, with 0.898 for the positive emotions items and 0.88 for the negative emotions items.

**Satisfaction with basic psychological needs.**   The appraisal of basic psychological needs—autonomy, competence, and relatedness—guided by self-determination Theory [58, 59] was central to this component. The study harnessed the tailored-for-elderly Basic Psychological Needs Satisfaction Scale (BPNSS) to probe these psychological cornerstones [60, 61]. This instrument comprises 18 items, each assessed using a 5-point Likert scale, and indicates the basic psychological needs of elderly people that are crucial for their psychological well-being. Within our study, the scale demonstrated significant reliability, with Cronbach's α coefficients of 0.86, 0.83, and 0.82 for autonomy, competence, and relatedness, respectively. These robust figures coalesced into a composite reliability score of 0.93 for the entire scale, underscoring its credibility in evaluating the intrinsic motivations and overall psychological well-being of elderly individuals.

## Results

Data collection was led by the first and second authors, who employed the intention-to-treat principle and multiple imputation methods. The analysis was performed using SPSS 27, with a predefined significance level of $p < 0.05$. A total of 230 surveys were distributed across two nursing homes in Yunhe County, Lishui City, Zhejiang Province. Following the analysis, 38 elderly residents from each nursing home, totaling 76 individuals, met the established inclusion criteria and voluntarily agreed to participate in the study (e.g., see S2 Fig). During the research phase, two participants from both the intervention and control groups opted out for health-related reasons. Notably, none of the participants underwent medical or psychotherapeutic treatment for depression or cognitive impairment during the intervention.

## Sociodemographics

Table 1 presents the detailed sociodemographic characteristics of the study participants. The intervention group had an average age of 78.34±4.83 years, whereas the average age of the control group was 82.63±4.98 years, indicating a statistically significant difference in age. A unique demographic characteristic of the nursing homes in Yunhe County is the preponderance of male elderly residents. Remarkably, one of the nursing homes, chosen at random, only had male participants who satisfied the inclusion benchmarks. This particularity substantially influenced the noticeable sex imbalance across groups. Both age and sex were incorporated as covariates in subsequent analytical models to counteract these differences, guaranteeing thorough comparisons. Apart from these aspects, no considerable differences were identified in other demographic characteristics or in the baseline scores of cognitive function or psychological health between the groups (p values > 0.05), highlighting the inherent comparability of the groups.

We specifically calculated the ICCs for the cognitive scores and mental health-related scores to comprehensively evaluate the potential intra-cluster variability in our CRCT. We found that the ICC values (ICC = 0.007) approached zero, explicitly indicating that the similarity among individuals within clusters was negligible [50–52]. This finding prompted us to employ traditional repeated-measures ANOVA to delve deeper into the changes in cognitive scores and

**Table 1. Baseline characteristics of elderly participants in the intervention and control groups.**

| Variables | Intervention group (n = 38) | Control group (n = 38) | p value | 95% CI |
|---|---|---|---|---|
| **Age (years), mean±SD** | 78.34±4.83 | 82.63±4.98 | 0.002 | [-10.53, -5.95] |
| **Sex** | | | 0 | [-0.79, -0.47] |
| Male | 38 (100%) | 24 (63%) | | |
| Female | 0 | 14 (37%) | | |
| **Education** | | | 0.26 | [-0.48, 0.06] |
| Primary school and no degree | 22 (58%) | 18(47%) | | |
| Middle school | 16 (42%) | 16 (42%) | | |
| High school or polytechnic school | 0 | 4 (11%) | | |
| **Occupation** | | | 0.1 | [-1.16, 0.01] |
| Technical personnel | 2 (5%) | 4 (11%) | | |
| Government agencies | 1 (3%) | 5 (13%) | | |
| Industrial workers | 1 (3%) | 2 (5%) | | |
| Farmers | 29 (76%) | 22 (58%) | | |
| Household affairs/home duties | 5 (13%) | 5 (13%) | | |
| **Marital status** | | | 0.36 | [-0.12, 0.33] |
| Widowed | 19 (50%) | 23 (60%) | | |
| Unmarried | 19 (50%) | 15 (40%) | | |

mental health-related indicators between the intervention and control groups from baseline to postintervention. Throughout this process, we considered time (i.e., baseline vs. postintervention) as the repeated measure factor and group (i.e., intervention vs. control) as a between-subjects factor. Given the significant age and sex differences between the two groups, age, and sex were incorporated as covariates in the ANOVA to account for potential influences. Notably, before conducting these pivotal analyses, we ensured that the data adhered to the assumptions of normality and homogeneity of variance.

## Primary outcome: Cognitive function

Table 2 presents the MoCA scores for both the intervention and control groups from baseline to postintervention.

No significant main effects were detected for time, $F_{(1, 72)} = 0.09$, $p > 0.05$, group, $F_{(1, 72)} = 0.34$, $p > 0.05$; age, $F_{(1, 72)} = 1.31$, $p > 0.05$; or sex, $F_{(1, 72)} = 0.92$, $p > 0.05$. Additionally, the effects of the interactions between time and age, $F_{(1, 72)} = 0.21$, $p > 0.05$, as well as between time and sex, $F_{(1, 72)} = 0.82$, $p > 0.05$, were not statistically significant. Nevertheless, a prominent interaction was observed between time and group, $F_{(1, 72)} = 18.23$, $p < 0.001$,

**Table 2. Distribution of MoCA scores between the intervention and control groups.**

| MoCA | Intervention group (M±SD) | Control group (M±SD) | t value | p value | 95% CI |
|---|---|---|---|---|---|
| **Baseline** | 13.11±4.07 | 15.53±6.65 | -1.91 | 0.08 | [-4.95, 0.11] |
| **Postintervention** | 16.29±5.03 | 14.37±7.14 | 1.36 | 0.179 | [-0.91, 4.75] |
| **t value** | -5.16 | 2.6 | | | |
| **p value** | 0 | 0.01 | | | |
| **95% CI** | [-4.44, -1.93] | [0.26, 2.06] | | | |

Abbreviations: MoCA, Montreal Cognitive Assessment.

with a partial $\eta^2$ of 0.23. Specifically, the intervention group exhibited a marked improvement in the average score for cognitive function, $t(37) = -5.16$, $p < 0.001$, Cohen's $d = -0.84$, 95% CI [-4.44, -1.93], whereas a significant decrease was observed in the control group, $t(37) = 2.6$, $p = 0.01$, Cohen's $d = 0.422$, 95% CI [0.26, 2.06]. These findings are illustrated in Fig 1.

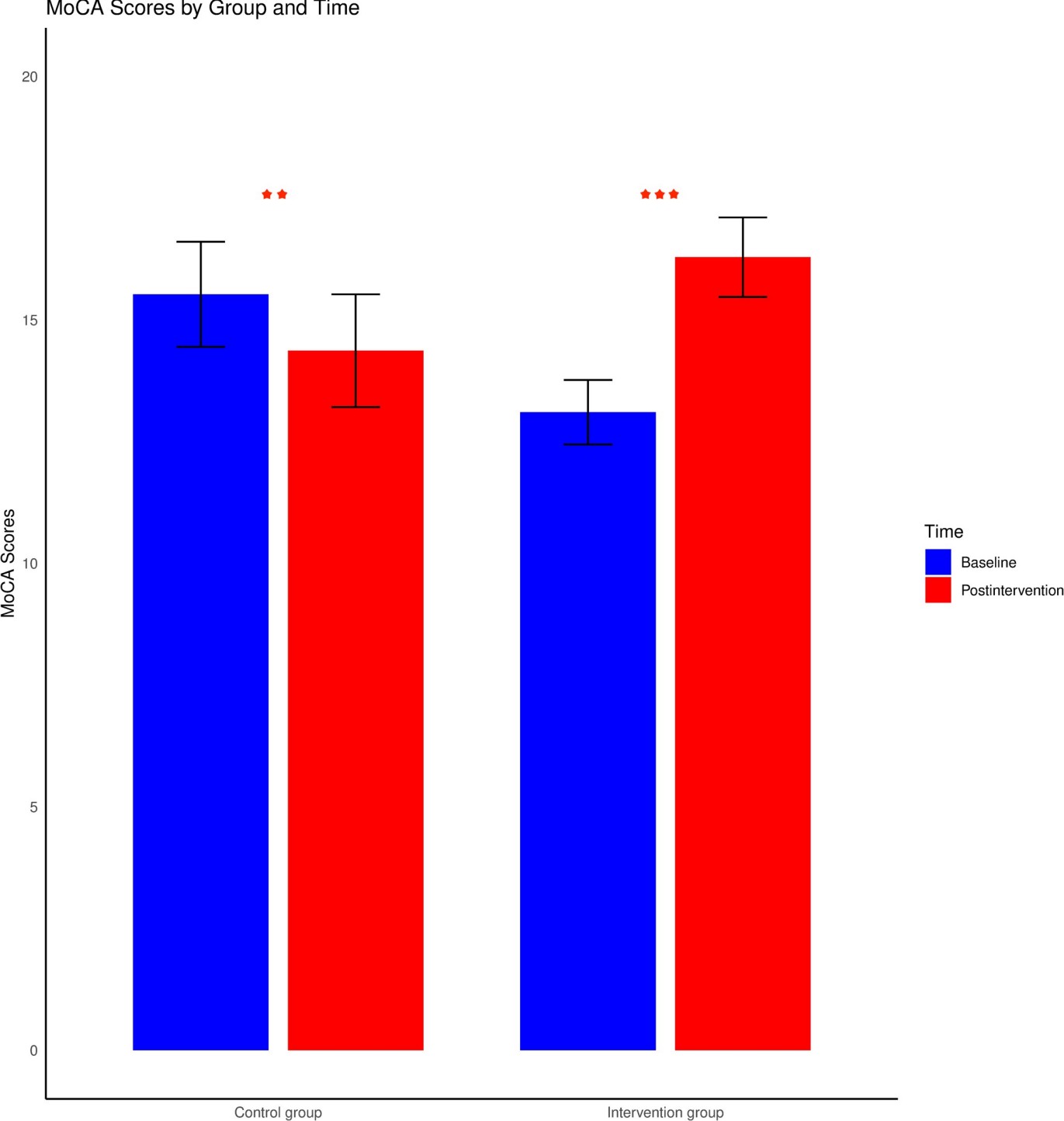

**Fig 1. Changes in MoCA scores from baseline to postintervention for both the intervention and control groups.** The error bars reflect the standard errors of the means. *** represents $p < 0.001$; ** represents $p < 0.01$; and * represents $p < 0.05$.

Furthermore, within the MoCA questionnaire, the intervention group displayed significant increases in scores for four items—namely, naming, language, memory, and orientation (e.g., see S1 Table). The score for the abstract reasoning item marginally significantly improved, with p = 0.09, indicating the efficacy of wooden toy training in enhancing certain cognitive abilities. No substantial differences were observed in the pre- and posttest scores of the intervention group for the remaining items: visuospatial/executive function and attention. Moreover, the control group exhibited a significant decrease in the attention domain, t (37) = 2.65, p = 0.01, 95% CI [0.35, 1.29], with no significant differences observed for the other items, p values > 0.05.

### Second outcome: Psychological well-being

**Emotional health.** Consistent with the analytical approach adopted for the MoCA scores, a repeated measures ANOVA was applied to evaluate various facets of emotional health. This assay included an analysis of depressive symptoms, as evidenced by the GDS-15 scores (see Table 3), and daily emotional experiences, as represented by scores pertaining to both positive and negative emotions.

The results of the ANOVA for the GDS-15 scores were as follows: neither the main effects of time or sex nor the effects of interactions between time and age, time and sex, or time and group reached statistical significance (p > 0.05). However, a significant main effect of age was observed, F (1, 72) = 4.91, p < 0.05, implying that depressive symptoms intensify with age. Additionally, the main effect of group was significant, F (1, 72) = 5.02, p < 0.05. Despite the nonsignificant interaction between time and group, after 8 weeks of wooden toy training, the intervention group presented a significant decrease (M = 7.18, 95% CI [0.35, 2.44]) in depression scores relative to the baseline scores, t (37) = 2.71, p < 0.05, with Cohen's d value of 0.447. Conversely, the control group displayed no notable change (95% CI [-0.35, 0.87]). However, both groups exhibited a reduction in their GDS-15 scores. These findings suggest the efficacy of wooden toy training in ameliorating depressive symptoms among elderly individuals with cognitive impairment. These findings are illustrated in Fig 2.

Of the 15 items assessed, five showed a statistically significant reduction in scores for the intervention group. Specifically, items such as "Do you feel pretty worthless the way you are now?", "Have you dropped many of your activities and interests?", "Do you feel you have more problems with memory than most?", "Do you feel your current situation is hopeless?", and "Do you feel energetic?" were statistically significant (p values < 0.05). The remaining items were not significantly different (p values > 0.05). In the control group, no items were significant different.

The results of the ANOVA of the scores for positive and negative emotions are as follows: in the context of positive emotions, no notable significant differences were detected. Concerning negative emotions, the main effects of time, sex, and group were not statistically significant

**Table 3. Distribution of GDS-15 scores between the intervention and control groups.**

| GDS-15 | Intervention group (M±SD) | Control group (M±SD) | t value | p value | 95% CI |
|---|---|---|---|---|---|
| Baseline | 8.58±2.83 | 8.53±3.33 | -0.14 | 0.89 | [-1.36, 1.46] |
| Postintervention | 7.18±2.1 | 8.26±3.04 | -1.91 | 0.08 | [-2.27, 0.11] |
| t value | 2.71 | 0.87 | | | |
| p value | 0.01 | 0.4 | | | |
| 95% CI | [0.35, 2.44] | [-0.35, 0.87] | | | |

Abbreviations: GDS-15, Geriatric Depression Scale.

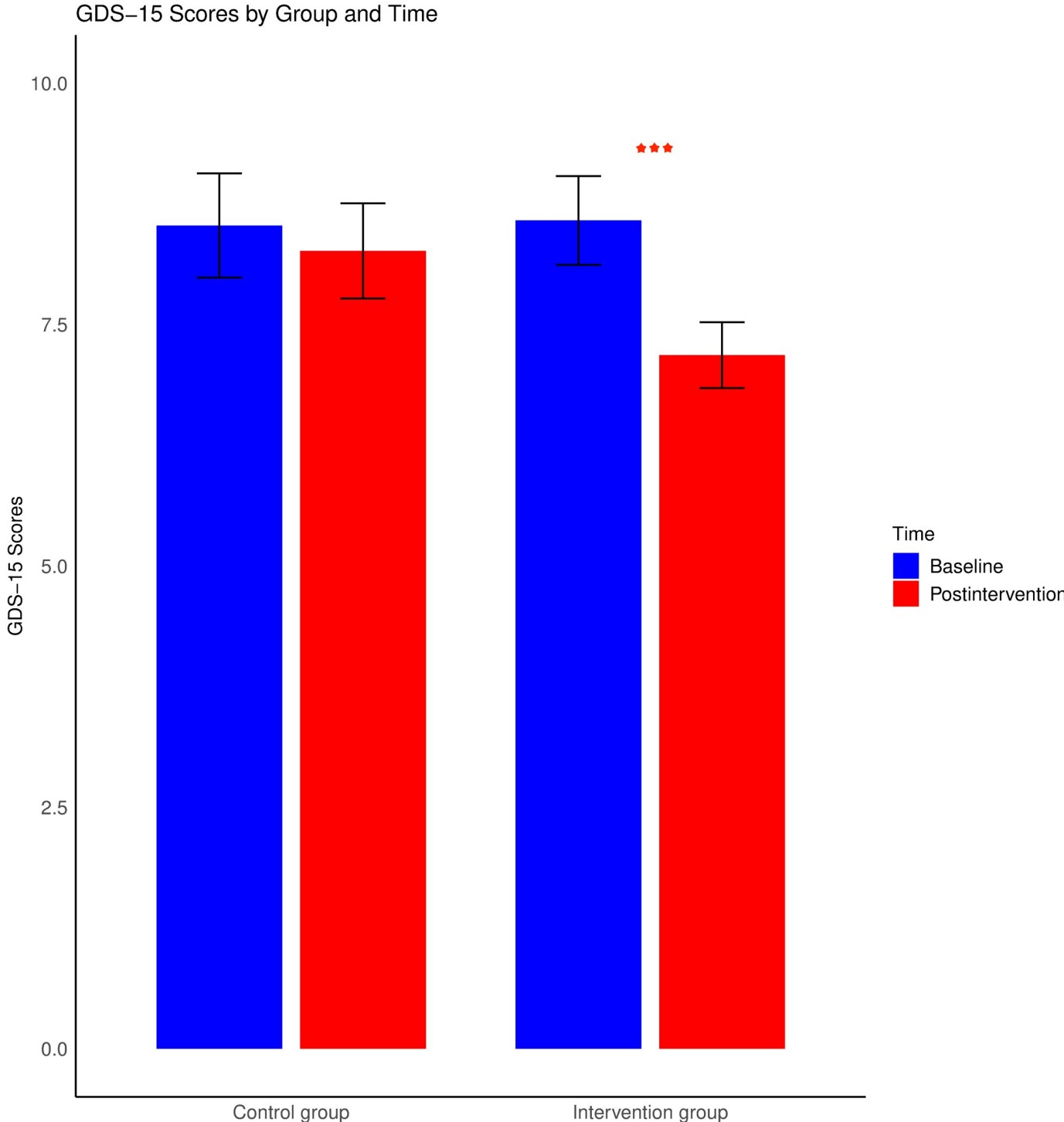

**Fig 2. Changes in GDS-15 scores from baseline to postintervention for both the intervention and control groups.** The error bars reflect the standard errors of the means. *** represents p < 0.001; ** represents p < 0.01; and * represents p < 0.05.

**Table 4. Distribution of BPNS scores between the intervention and control groups.**

| BPNS | BPNS_A | | | | BPNS_R | | | | BPNS_C | | | |
|---|---|---|---|---|---|---|---|---|---|---|---|---|
| | Intervention group (M±SD) | Control group (M±SD) | t value | p value | Intervention group (M±SD) | Control group (M±SD) | t value | p value | Intervention group (M±SD) | Control group (M±SD) | t value | p value |
| **Baseline** | 17.74±5.74 | 19.92±5.75 | -1.66 | 0.1 | 20.32±5.28 | 22.16±4.46 | -1.64 | 0.11 | 16.29±5.07 | 18.34±4.26 | -2.88 | 0.06 |
| **Postintervention** | 19.37±5.66 | 19.05±5.45 | 0.25 | 0.81 | 22.95±5 | 22.08±4.04 | 0.072 | 0.94 | 20±4.9 | 18.84±5.01 | 1.02 | 0.31 |
| **t value** | -1.49 | 1.67 | | | -2.54 | 0.13 | | | -3.41 | 1.562 | | |
| **p value** | 0.15 | 0.1 | | | 0.02 | 0.9 | | | 0.002 | 0.127 | | |
| **95% CI** | | | | | [-4.73, -0.53] | | | | [-5.92, -1.51] | | | |

(p > 0.05). Similarly, interaction effects involving time and age, time and sex, and time and group did not exhibit significance (p > 0.05). However, a pronounced main effect of age was present, $F_{(1, 71)} = 25.35$, $p < 0.001$, indicating an increased intensity of negative emotions with increasing age.

Although the time–group interaction did not reach significance, further analyses within the intervention group indicated a statistically significant reduction in negative affective scores following an 8-week wooden toy intervention. This reduction was most evident in emotions such as "unease, pain, irritability, hostility, and depression". Comparable changes were not observed in the control group (e.g., see S2 Table). These results highlight that 8 weeks of wooden toy training can notably reduce certain negative emotions, even if the impact is not consistent across all categories of negative emotions.

**Basic psychological needs satisfaction.** Table 4 delineates basic psychological needs satisfaction (BPNS) among elderly individuals, which was segmented into three dimensions: autonomy (BPNS_A), relatedness (BPNS_R), and competence (BPNS_C). A multivariate analysis of variance (MANOVA) was employed to discern potential shifts in satisfaction scores across these facets between the intervention and control groups both before and after the intervention.

The results revealed that the main effects of group, age, and sex were not statistically significant (p > 0.05). However, a significant main effect of time was detected, $F_{(3, 144)} = 5.88$, $p = 0.001$, $\eta^2 = 0.11$, indicating a marked shift in the satisfaction of needs before and after the intervention. Neither the interactions between age and time nor those between sex and time reached significance (p values > 0.05). Crucially, a significant interaction between group and time was evident, $F_{(3, 144)} = 4.31$, $p < 0.01$, $\eta^2 = 0.0$. A subsequent univariate ANOVAs revealed a significant main effect of time, $F_{(1, 146)} = 7.18$, $p < 0.01$, $\eta^2 = 0.05$, and a significant interaction effect between group and time for the competence satisfaction variable, $F_{(1, 146)} = 4.17$, $p < 0.05$, $\eta^2 = 0.03$. Specifically, after 8 weeks of wooden toy training, the intervention group presented a pronounced increase in competence need satisfaction, $t_{(75)} = -3.41$, $p < 0.01$, M = 20, 95% CI [-5.92, -1.51], Cohen's d = -0.55 relative to the baseline scores (M = 16.29) (see Fig 3). Additionally, a significant increase in relatedness need satisfaction was observed within the intervention group, $t_{(75)} = -2.54$, $p < 0.05$, M = 22.95, 95% CI [-4.73, -0.53], Cohen's d = -0.31 relative to the baseline scores (M = 20.32) (see Fig 4). Neither main effects nor interaction effects were observed for scores for autonomy need satisfaction. The control group did not exhibit any significant differences.

## Discussion

The current study aimed to investigate the impact of wooden toy training on mitigating cognitive decline in elderly individuals with cognitive impairment, with a specific focus on assessing

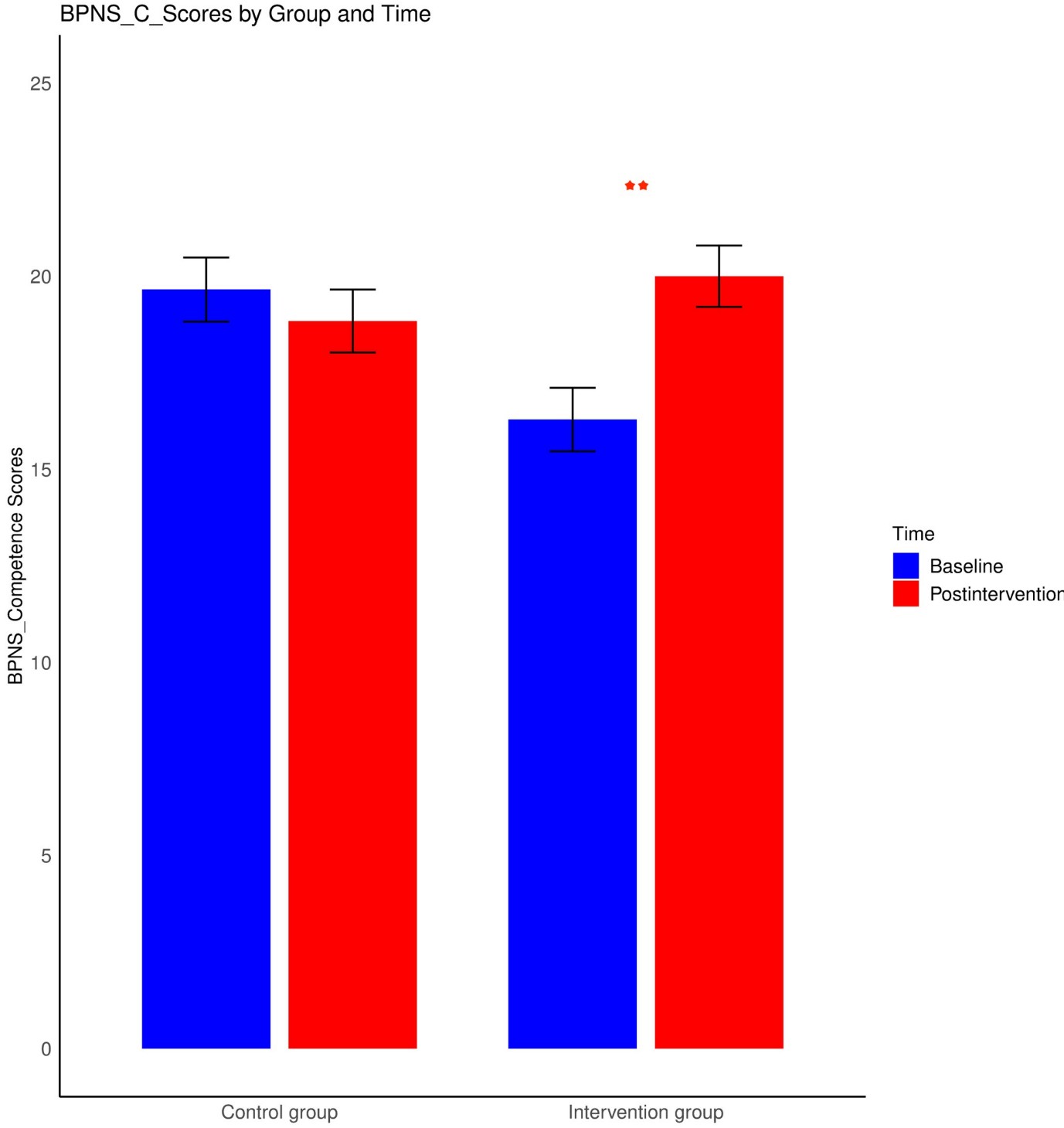

**Fig 3. Changes in BPNS_Competence scores from baseline to postintervention for both the intervention and control groups.** The error bars reflect the standard errors of the means. *** represents p < 0.001; ** represents p < 0.01; and * represents p < 0.05.

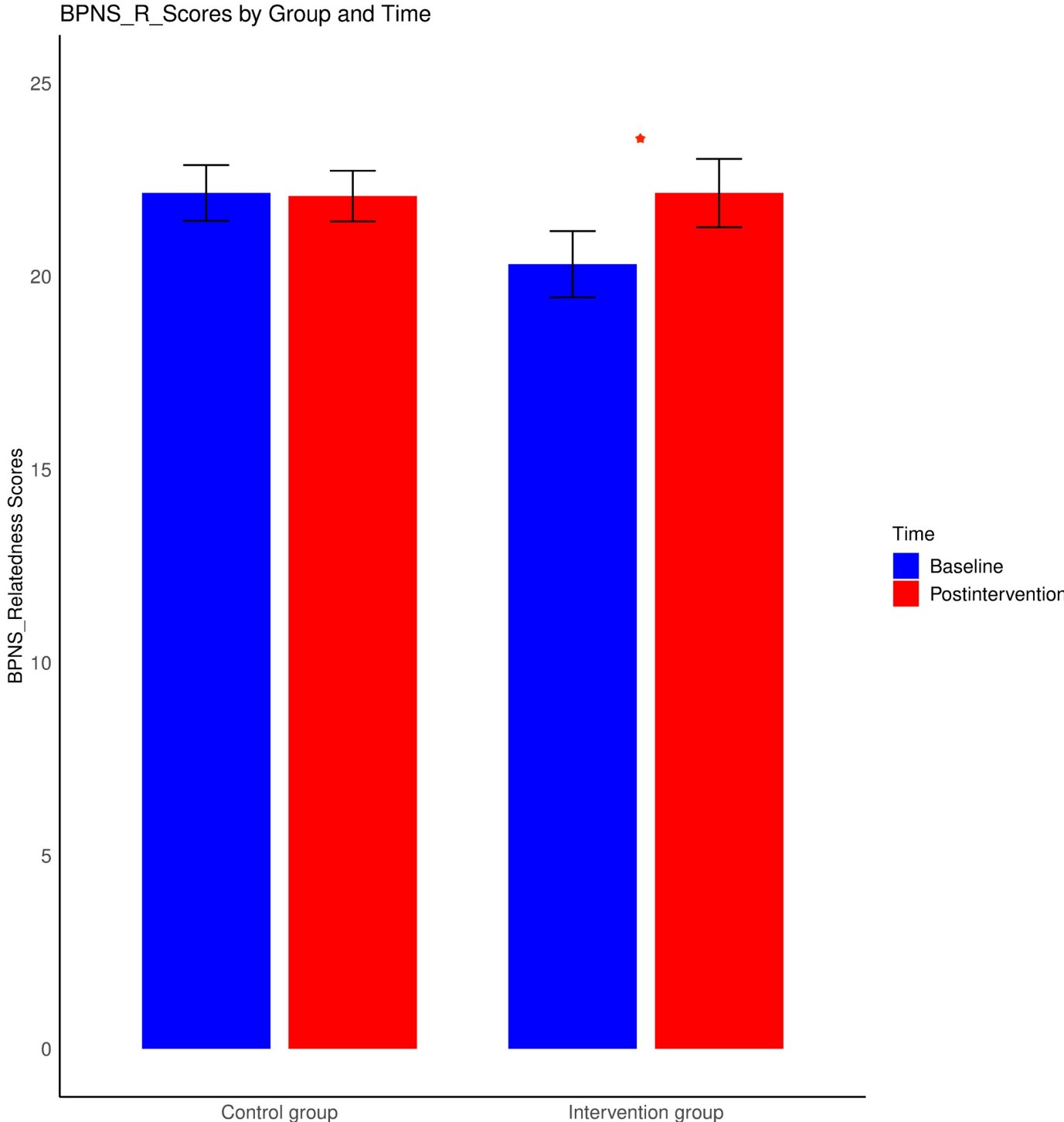

**Fig 4. Changes in BPNS_Relatedness scores from baseline to postintervention for both the intervention and control groups.** The error bars reflect the standard errors of the means. *** represents $p < 0.001$; ** represents $p < 0.01$; and * represents $p < 0.05$.

how this training affects cognitive functions and psychological well-being, and exploring whether the enhancement of cognitive skills and mental health contributes to slowing or reversing the progression of cognitive decline. Our findings suggest that wooden toy training can significantly bolster the cognitive abilities of elderly individuals while diminishing the symptoms of depression and certain negative emotional experiences. Moreover, such wooden toy training appears to greatly satisfy the needs for competence and relatedness in older individuals. Our findings not only align with the initially set objectives but also provide additional insights into the role of wooden toy training in fulfilling the psychological and social needs of elderly individuals. Our research provides compelling evidence for the potential benefits of wooden toys. These findings indicate that wooden toy training effectively enhances the cognitive abilities of elderly individuals. This outcome aligns with previous studies that highlighted the benefits of cognitive stimulation in slowing cognitive decline among older adults [40, 62–64]. We observed significant improvements in specific domains, including naming, language, memory, abstract reasoning, and orientation. This specificity of improved cognitive functioning suggests that certain cognitive domains are more responsive to the cognitive stimulation provided by wooden toys. This sentiment resonates with findings that not all cognitive domains benefit equally from cognitive exercises [65].

Wooden toys, through their sensory–motor interactions and cognitive challenges, seem to activate and reinforce neural pathways in the brain. For example, manipulating a wooden puzzle demands spatial cognition, whereas shape-sorting toys require fine motor skills. This engagement activates brain regions essential for motor coordination and problem solving, subsequently enhancing various cognitive functions. In terms of the broader mechanism, continuous engagement with wooden toys could promote neural plasticity. This inherent capacity of the brain to form and reorganize synaptic connections is crucial for adapting to new information and experiences [43]. Given this idea, such plasticity might explain why wooden toys provide cognitive benefits. The noted benefits in areas such as naming, language, memory, and orientation indicate that wooden toys may rejuvenate or fortify neural connections, reviving pathways that might have diminished efficiency with age [66]. Given these findings and the supporting literature, there's potential for tailoring wooden toy interventions could be tailored to target specific cognitive domains. This approach suggests that we could create personalized therapeutic strategies. Depending on an individual's cognitive needs, specific wooden toys can be selected and customized to provide the best cognitive stimulation.

Elderly individuals with cognitive impairment often experience heightened depressive symptoms [67, 68]. The results of our study illuminate the potential of wooden toy training to address this concern. After an 8-week training regimen, participants exhibited not only a marked decrease in traditional depressive indicators, such as feelings of worthlessness and diminished interest in activities (as indicated by GDS-15 scores), but also broader negative emotional experiences including "unease, pain, irritability, hostility, and depression".

The intrinsic qualities of wooden toys appear to play a pivotal role in this process. Engaging with wooden toys provides a tactile, immersive experience that transcends mere cognitive stimulation. Sensory engagement with wood, a natural material, may elicit emotional comfort [69], reflecting the biophilia hypothesis of Wilson (1984) [70] which postulates a deep-rooted human affinity with nature. Such interactions with natural materials have been suggested to exert therapeutic effects, including mood improvement and stress reduction [32, 34]. Moreover, wooden toys, reminiscent of a bygone era, might evoke feelings of nostalgia in elderly individuals. Nostalgia has been posited to counteract loneliness, boredom, and anxiety, serving as a buffer against negative emotions [71]. This dual impact of wooden toys—cognitive engagement combined with an emotional sanctuary—harkens back to the broader understanding of how activities can combat feelings of isolation and depression in older adults [72].

Our findings echo those of other studies that emphasize the role of meaningful activities in guarding elderly people against depressive tendencies [73, 74]. Zhang et al. [75] further argue for interventions that are participatory, culturally resonant, and group-based, suggesting that these interventions are especially effective against geriatric depression. Similar to the insights of Carstensen & Chi [76], our study highlights the notion that while cognitive abilities might wane with age, emotionally geared interventions can indeed preserve, if not enhance, emotional stability in elderly individuals.

In this study, we also found that elderly participants in the intervention group experienced a significant improvement in their sense of competence and a marginal increase in their feelings of relatedness. Engaging with wooden toys seemed to bolster their self-efficacy and sense of control, potentially enhancing their overall cognitive and emotional well-being. Our findings resonate with earlier research, underscoring the potential of games and leisure activities in enhancing self-efficacy and a sense of control among the elderly [77–79], especially in frail older adults [80, 81]. Additionally, the collaborative nature of certain activities involving wooden toys might have enhanced social interactions among participants, thereby increasing their feelings of relatedness. This observation aligns with the findings of Chen [82], who highlighted the crucial role of social connections derived from leisure activities in increasing the well-being of elderly individuals. Such collaborative endeavors not only promote cognitive and emotional health but also emphasize the inherent value of relatedness in aging populations. However, our findings also suggest that wooden toys, while beneficial, are not a universal solution. They appeared to have a positive effect on certain aspects of the participants' psychological state, but not all aspects. This results is consistent with the understanding that individual responses to interventions can vary and that not all emotional or psychological challenges can be addressed with a singular approach [58]. Hence, our results not only emphasize the therapeutic potential of wooden toys for the elderly but also align with a broader academic consensus on the positive impact of games and recreational pursuits on the well-being of elderly individuals.

## Limitations

Our study illuminates the promising therapeutic potential of wooden toy interventions in geriatric care, particularly in the context of the improving the cognitive and emotional functioning of elderly individuals with cognitive deficits. However, it is not without its limitations.

First, the age and sex disparities between the intervention and control groups are our central concerns. Despite controlling for these variables in our analysis, the age discrepancy and sex imbalance—in which the intervention cohort exclusively comprised males—remain potential sources of bias. Sex significantly affects cognitive processing, emotional responses, and receptivity to interventions, emphasizing the need for sex representation in future studies for broader applicability, especially for female populations.

Second, wooden toys are noted for their portability, familiarity, and affordability. These inherent attributes make them particularly accessible and appealing to elderly individuals, promoting their integration into daily routines. Their intuitive nature fosters prolonged engagement, suggesting that the therapeutic benefits might be more enduring than fleeting. Nevertheless, the duration of these benefits still requires rigorous examination. Continuous assessments are pivotal to discern whether the improvements obtained from wooden toy training are sustained or diminish over time.

Third, our study employed a diverse range of wooden toys, each introducing unique cognitive and emotional challenges. A more granular, toy-specific analysis is essential for decoding each toy's distinct contribution, paving the way for tailored therapeutic interventions.

Fourth, the cultural and geographical specificity of our study, rooted in a Chinese context, introduces questions about the universality of our findings. While wooden toys are deeply embedded in Chinese traditions, their broader efficacy and resonance in diverse cultural settings remain uncharted.

Finally, beyond the tangible benefits, the quest to understand the underlying mechanisms remains. Is the therapeutic efficacy primarily derived from cognitive stimulation, or do emotional and sensory interactions with the toys wield a stronger influence? Distilling these nuances will further optimize the therapeutic utilization of wooden toys.

As the global demographic has evolved with a burgeoning elderly population, culturally attuned and accessible interventions such as wooden toys have gained prominence. Future studies must adopt sex-balanced and age-homogeneous cohorts to fully harness their potential of these interventions, providing comprehensive insights into their therapeutic efficacy. With such informed strategies, we can better enrich the lives of elderly people, intertwining innovation with cultural sensitivity.

## Conclusions

With the global trend of increasing age, cognitive impairments in elderly individuals have become increasingly concerning, deeply affecting their mental health and overall well-being. When addressing this pressing issue, innovative and effective interventions tailored to this demographic are urgently needed. Considering this need, our study delved into the therapeutic potential of traditional "wooden toy" training, aspiring to discern its efficacy in counteracting cognitive decline and bolstering psychological well-being among elderly individuals.

Our results are promising, revealing that this culturally rooted intervention can significantly enhance cognitive ability, mitigate depressive symptoms and other negative emotional experiences, and cater to fundamental psychological needs. Importantly, these methods not only aid in mitigating the impact of cognitive decline, thereby improving the quality and effectiveness of elderly care in nursing homes but also have the potential to benefit societal welfare. This study underscores the importance of leveraging such traditional methods in addressing the burgeoning challenge of cognitive impairment. This approach not only introduces a novel, effective, and cost-efficient nonpharmacological therapy but also promises a more enriched and fulfilling life for elderly individuals facing cognitive challenges.

## Supporting information

**S1 Fig. Game examples in wooden toy training.**
(PNG)

**S2 Fig. Recruitment and flow diagram for cognitive impairment study.**
(PNG)

**S1 Table. Distribution of MoCA item scores for intervention and control groups.**
(DOCX)

**S2 Table. Negative affective scores from baseline to postintervention.**
(DOCX)

## Acknowledgments

We would like to express our profound gratitude to the medical staff at the Fuyun Yuanhe Street Community Health Service Center in Yunhe County. Their invaluable assistance in facilitating data collection at the Nursing Homes was instrumental to this study.

## Author Contributions

**Formal analysis:** Qiuping Cheng.

**Funding acquisition:** Qiuping Cheng, Lu Li.

**Investigation:** Qiuping Cheng, Hanqian Wang, Mengni Cui.

**Methodology:** Qiuping Cheng.

**Project administration:** Qunlong Wang, Lu Li.

**Supervision:** Qunlong Wang, Lu Li.

**Visualization:** Qiuping Cheng.

**Writing – original draft:** Qiuping Cheng.

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
