## [Decision Letter · Decision Letter 0]

7 Aug 2024

PONE-D-24-23952Efficacy of Wooden Toys Training on Alleviating Cognitive Decline in Elderly Individuals with Cognitive Impairment: A Cluster Randomized Controlled StudyPLOS ONE

Dear Dr. Wang,

Thank you for submitting your manuscript to PLOS ONE. After careful consideration, we feel that it has merit but does not fully meet PLOS ONE’s publication criteria as it currently stands. Therefore, we invite you to submit a revised version of the manuscript (MINOR REVISION) that addresses the points raised during the review process.

Please submit your revised manuscript in Sep 21 2024 11:59PM. If you will need more time than this to complete your revisions, please reply to this message or contact the journal office at plosone@plos.org. Please include the following items when submitting your revised manuscript:A rebuttal letter that responds to each point raised by the academic editor and reviewer(s). You should upload this letter as a separate file labeled 'Response to Reviewers'.A marked-up copy of your manuscript that highlights changes made to the original version. You should upload this as a separate file labeled 'Revised Manuscript with Track Changes'.An unmarked version of your revised paper without tracked changes. You should upload this as a separate file labeled 'Manuscript'.If applicable, we recommend that you deposit your laboratory protocols in protocols.io to enhance the reproducibility of your results. Protocols.io assigns your protocol its own identifier (DOI) so that it can be cited independently in the future. For instructions see: https://journals.plos.org/plosone/s/submission-guidelines#loc-laboratory-protocols. Additionally, PLOS ONE offers an option for publishing peer-reviewed Lab Protocol articles, which describe protocols hosted on protocols.io. Read more information on sharing protocols at https://plos.org/protocols?utm_medium=editorial-email&utm_source=authorletters&utm_campaign=protocols.

We look forward to receiving your revised manuscript.

Kind regards,

Vaitsa Giannouli

Academic Editor

PLOS ONE

Journal Requirements:

Reviewers' comments:

Reviewer's Responses to Questions

**Comments to the Author**

1. Is the manuscript technically sound, and do the data support the conclusions?

Reviewer #1: Yes

Reviewer #2: Yes

2. Has the statistical analysis been performed appropriately and rigorously? 

Reviewer #1: Yes

Reviewer #2: Yes

3. Have the authors made all data underlying the findings in their manuscript fully available?

Reviewer #1: No

Reviewer #2: Yes

4. Is the manuscript presented in an intelligible fashion and written in standard English?

Reviewer #1: Yes

Reviewer #2: Yes

5. Review Comments to the Author

Reviewer #1: This is an interesting paper, well-designed and well-written. I have several comments regarding its improvment:

You should quote the source for the tool: "Cognitive functions were assessed using the Chinese adaptation of the Montreal Cognitive Assessment (MoCA), constituting the study's main outcome measure", "It is particularly applicable for identifying MCI, with scores below 26 typically indicating potential cognitive decline."

Regarding "The scale demonstrated a strong internal consistency with a Cronbach's alpha of 0.818.", is it a result from your study or you should quote a source?

You should quote the source for the tool: "We utilized the Geriatric Depression

Scale (GDS-15) for its prowess in gauging depressive tendencies in the elderly. Scores span 0

to 15, with those exceeding 8 signaling emerging depressive symptoms, underscoring an

escalated risk of depression. The scale's reliability is underscored by Cronbach's α of 0.82."

The source Wilson (1984) quoted in the text is missing in References.

The reviewer has not found any way to access the data, no matter of the authors' statement "Yes - all data are fully available without restriction" The authors should describe how the data could be accessed.

Reviewer #2: This article focuses upon the application of an intervention in which wooden toys are used in elderly in order to eliminate the deficits associated with MCI. The introduction addresses the need for the current research. The research design is sound as well as the statistical analysis followed data collection. Finally, the discussion summarizes the main findings which are associated with previous research. Also, limitations are elaborated as well as the conclusions of the research findings. Nevertheless, it should be given attention to the list of references. Referencing does not follow the 7th edition of APA and therefore it should be revised.

6. PLOS authors have the option to publish the peer review history of their article (what does this mean?). If published, this will include your full peer review and any attached files.

Reviewer #1: **Yes: **Stanislava Stoyanova

Reviewer #2: No

---

## [Author Response · Author response to Decision Letter 0]

14 Aug 2024

Dear Dr. Giannouli and Esteemed Reviewers,

I hope this message finds you well. We are writing to submit our revised manuscript entitled “Efficacy of wooden toys training on alleviating cognitive decline in elderly individuals with cognitive impairment: a cluster randomized controlled study” for further consideration in PLOS ONE. We are deeply grateful for the detailed reviews and insightful comments provided by each of the reviewers, as well as your guidance as our academic editor. These contributions have been invaluable in enhancing the overall quality and clarity of our manuscript.

We have thoroughly addressed each point raised during the review process and have made careful revisions accordingly. Below, we provide a detailed point-by-point response to all the feedback received, illustrating the changes made to our manuscript and offering clarifications where necessary.

Response to Point 1:

Thank you for directing us to the PLOS ONE style templates. We have thoroughly reviewed the provided templates and have made careful revisions to ensure our manuscript fully complies with PLOS ONE’s style requirements. These modifications include adjustments to file naming and formatting as specified in the guidelines. The details of these changes are clearly marked in the “Revised Manuscript with Track Changes” for easy identification. A clean version of the manuscript has also been uploaded for your review.

Response to Point 2:

In response to your recommendation for thorough copyediting, we have engaged the services of The American Journal Experts (AJE) to edit our manuscript for language usage, spelling, and grammar. The specific changes made by AJE can be seen in the “Revised Manuscript with Track Changes” file, where all modifications have been highlighted for easy review. We believe these efforts have significantly enhanced the clarity and readability of our manuscript.

Response to Point 3:

Thank you for highlighting the discrepancies between the ‘Funding Information’ and ‘Financial Disclosure’ sections. We have thoroughly reviewed and synchronized these sections to ensure consistency and accuracy. 

The corrected ‘Funding Information’ at the end of the manuscript is as follows:

“Funding Information

Receiver: Qiuping Cheng; Fund Name: The Planning Projects of Zhejiang Provincial Philosophy and Social Sciences (24NDJC193YB); Open Fund Project of the Modern Service Research Center of Zhejiang Shuren University (SXFJZ202301). These funds were utilized in the data collection process.

Receiver: Lu Li; Fund Name: The National Social Science Fund of China (21BGL235). This funding supported the research design and research survey.”

The detailed revisions to the ‘Funding Information’ are documented in both the ‘Revised Manuscript with Track Changes’. We believe these amendments ensure transparency and comply with the requirements for funding disclosure in academic publishing. Additionally, we have included an updated statement regarding these changes in our cover letter, specifically highlighted for clarity and transparency. Please refer to the highlighted section in the cover letter for further details.

Response to Point 4:

Thank you for your guidance regarding the placement of the ethics statement within our manuscript. We have carefully revised the document and now clearly include the ethics statement exclusively in the first paragraph of the Methods section, as you suggested. We have removed the ethics statement from the end of the manuscript to ensure compliance with the submission guidelines. Additionally, we have revised the Acknowledgements section, removing content that was redundant with the text and content that overlaps with the information required in the submission system. Specific details can be found at the end of the manuscript in the Acknowledgements section:

“Acknowledgements

We would like to express our profound gratitude to the medical staff at the Fuyun Yuanhe Street Community Health Service Center in Yunhe County. Their invaluable assistance in facilitating data collection at the Nursing Homes was instrumental to this study.”

The changes can be verified in the ‘Revised Manuscript with Track Changes’.

Response to Point 5:

Thank you for your instructions regarding the Supporting Information captions. Following your guidance and the journal’s Supporting Information guidelines, we have added captions for all Supporting Information files at the end of our manuscript. Below are the captions provided for each file:

Supporting Information

S1 Fig. Game Examples in Wooden Toys Training

S2 Fig. Recruitment and Flow Diagram for Cognitive Impairment Study

S1 Table. Distribution of MoCA item Scores for Intervention and Control Groups

S2 Table. Negative Affective Scores from Baseline to Post-Intervention

We have also updated the in-text citations accordingly to ensure consistency throughout the document. Specifically, these updates have been made in the following locations:

(1) In the Methods-Intervention program section, first paragraph, sixth line (see S1 Fig).

(2) In the Results section, first paragraph, sixth line (see S2 Fig).

(3) In the Results-Primary outcome: Cognitive function section, third paragraph, third line (see S1 Table).

(4) In the Results-Second outcome: Psychological well-being-Emotional Health section, fifth paragraph, fifth line (see S2 Table).

These adjustments can be reviewed in the ‘Revised Manuscript with Track Changes’.

Response to Point 6:

Thank you for your guidance on reviewing our reference list for completeness and accuracy. We acknowledge that it was an oversight on our part to have missed several key references initially mentioned by Reviewer #1. We have since carefully reviewed our reference list and cross-checked each citation against the manuscript text. We have now added the previously omitted citations and have updated the reference list accordingly to ensure that all references are correctly included and accurately cited. This update not only corrects our initial oversight but also enhances the overall integrity and scholarly rigor of our manuscript.

Additionally, we have conducted a detailed examination of each cited paper to determine if any have been retracted. We are pleased to report that none of the papers in our reference list have been marked as retracted in the academic databases we consulted, including PubMed, Web of Science, and Scopus. We have ensured that our manuscript adheres to the highest standards of academic integrity and accuracy.

Response to Reviewer #1:

Thank you for your constructive feedback and attention to detail. We have carefully considered your comments and have made the following updates to our manuscript:

Response to Point 1: Source for Cognitive Assessment Tools

Thank you for your comment regarding the citation of the tools used for cognitive assessment in our study. We have now cited the sources for the Chinese adaptation of the Montreal Cognitive Assessment (MoCA) as follows:

“Nasreddine ZS, Phillips NA, Bédirian V, Charbonneau S, Whitehead V, Collin I, et al. The Montreal Cognitive Assessment, MoCA: A Brief Screening Tool For Mild Cognitive Impairment. J Am Geriatr Soc. 2005;53: 695–699. doi:10.1111/j.1532-5415.2005.53221.x

Hong Y, Zeng X, Zhu CW, Neugroschl J, Aloysi A, Sano M, et al. Evaluating the Beijing Version of Montreal Cognitive Assessment for Identification of Cognitive Impairment in Monolingual Chinese American Older Adults. J Geriatr Psychiatry Neurol. 2022;35: 586–593. doi:10.1177/08919887211036182”

These references have been inserted in the manuscript in the section “Methods-Measures”, specifically in the second paragraph, third line from the end. We have also updated the References section accordingly to include these citations (Nos. 53 and 54).

We trust that these amendments satisfactorily address your concerns and enhance the accuracy and transparency of our report.

Response to Point 2: Internal Consistency of the Scale

Thank you for your valuable feedback. We have addressed the point about the source of the Cronbach’s alpha reported for the MoCA scale. The Cronbach’s alpha of 0.818 was indeed calculated from data collected in our study. To clarify this and ensure transparency, we have updated the text in the manuscript accordingly. This revision has been made in the Methods-Measures section, specifically in the second paragraph, last sentence, to clearly indicate that this reliability measure was derived from our current study’s data. 

The updated text now reads: “In our study, the MoCA scale showed good internal consistency, with a Cronbach’s alpha of 0.818 calculated from our data.” 

Furthermore, we have not only revised the reliability data for the MoCA scale but also reviewed and confirmed the reliability of other scales used in our study. Detailed revisions have been made to the Methods-Measures section, specifically in the fourth paragraph, lines five and six, lines eleven and twelve, as well as in the fifth paragraph, line seven. These updates further ensure that all scales used in our research meet the necessary reliability standards to support our findings.

Response to Point 3: Source for Geriatric Depression Scale (GDS-15)

Thank you for your comment regarding the citation of the tools used for depression assessment in our study. We have now cited the sources for Geriatric Depression Scale (GDS-15) as follows:

“Guerin JM, Copersino ML, Schretlen DJ. Clinical utility of the 15-item geriatric depression scale (GDS-15) for use with young and middle-aged adults. J Affect Disord. 2018;241: 59–62. doi:10.1016/j.jad.2018.07.038

Sheikh JI, Yesavage JA. Geriatric Depression Scale (GDS): Recent evidence and development of a shorter version. Clin Gerontol J Aging Ment Health. 1986;5: 165–173.”

These references have been inserted in the manuscript in the section “Methods-Measures,” specifically in the fourth paragraph, third line. We have also updated the References section accordingly to include these citations (Nos. 55 and 56).

Response to Point 4: Missing Reference

We appreciate your attention in identifying the omission in our references. It was indeed an oversight on our part that the citation for Wilson (1984) (see in the second paragraph of the Discussion section, fourteenth line from the end) was previously missing from our manuscript. We have now corrected this error and included the reference in the References section as follows:

70.Wilson EO. Biophilia: the human bond with other species. Cambridge, Mass.: Harvard Univ. Press; 1984.

Response to Point 5: Data Accessibility

Thank you for pointing out the lack of clear access instructions for our data. We acknowledge the oversight in our previous statement that did not adequately explain how the data can be accessed despite claiming full availability. To rectify this, we have updated the Data Availability Statement in our manuscript to provide explicit access details.

“Data Availability

The final dataset is available on the figshare, DOI: 10.6084/m9.figshare.26695081. If you encounter any issues accessing the data or require further assistance, please contact the corresponding author, Qunlong Wang, at the email: zjsrujzyl@126.com. 

We apologize for any confusion caused and have taken steps to ensure that all our data are accessible in accordance with ethical guidelines and journal policies. This amendment enhances the transparency and accessibility of our research.

Response to Reviewer #2:

Thank you very much for your detailed review and appreciation of our research.

Regarding your concerns about the referencing style, we have thoroughly reviewed our reference list and have revised it to adhere strictly to the PLOS ONE reference format. Additionally, we have identified and included five key references that were previously missing from our list. These updates ensure that our citations are both complete and formatted correctly according to the journal`s requirements.

We believe these revisions address your concerns and enhance the manuscript`s compliance with journal standards. Thank you once again for your valuable input, which has helped improve the quality and accuracy of our work.

We have carefully considered and responded to all the points you raised in the review. We are grateful for the insightful comments that have undoubtedly strengthened the quality of our manuscript. Thank you for the opportunity to improve our work. We appreciate your continued guidance and are eager to hear your further feedback.

Sincerely,

Qiuping Cheng

Ph.D.

Zhejiang Shuren University

August 13, 2024

---

## [Editor Report · Decision Letter 1]

16 Aug 2024

Efficacy of wooden toy training in alleviating cognitive decline in elderly individuals with cognitive impairment: a cluster randomized controlled study

PONE-D-24-23952R1

Dear Dr. Qiuping Cheng,

We’re pleased to inform you that your manuscript has been judged scientifically suitable for publication and will be formally accepted for publication once it meets all outstanding technical requirements.

Kind regards,

Vaitsa Giannouli

Academic Editor

PLOS ONE
---

## [Editor Report · Acceptance letter]

7 Oct 2024

PONE-D-24-23952R1 

PLOS ONE

Dear Dr. Wang, 

I'm pleased to inform you that your manuscript has been deemed suitable for publication in PLOS ONE. Congratulations! Your manuscript is now being handed over to our production team.

Kind regards, 

on behalf of

Dr. Vaitsa Giannouli 

Academic Editor

PLOS ONE